# Floral Developmental Morphology and Biochemical Characteristics of Male Sterile Mutants of *Lagerstroemia indica*

**DOI:** 10.3390/plants13213043

**Published:** 2024-10-30

**Authors:** Fuyuan Deng, Liushu Lu, Lu Li, Jing Yang, Yi Chen, Huijie Zeng, Yongxin Li, Zhongquan Qiao

**Affiliations:** 1Hunan Provincial Key Laboratory of Forest Clonal Breeding, Hunan Academy of Forestry, Changsha 410004, China; 2College of Life Science and Technology, Central South University of Forestry and Technology, Changsha 410004, China

**Keywords:** *Lagerstroemia indica*, male sterility, pollen development, active oxygen metabolism

## Abstract

Male sterility is a common phenomenon in higher plants and often plays an important role in the selection of superior offspring. ‘Xiang Yun’ is a mutant of *Lagerstroemia indica* that does not bear fruit after flowering, and its flowering period is significantly longer than that of normal *L. indica*. To explore the timing and molecular mechanisms of sterility in ‘Xiang Yun’, this study determined the period of sterility through anatomical observation and compared the content of nutrients and the activity of antioxidative enzymes at different stages of flower development. Finally, sequence alignment and qPCR were used to analyze the differences in pollen development genes between ‘Xiang Yun’ and ‘Hong Ye’. The results showed that the anthers of ‘Xiang Yun’ dispersed pollen normally, but the pollen grains could not germinate normally. Observations with scanning electron microscopy revealed that the pollen grains were uneven in size and shriveled in shape. Further observation of anther sections found that abnormal development of the microspores began at the S2 stage, with the callose wall between microspores of ‘Xiang Yun’ being thicker than that of ‘Hong Ye’. In addition, during the flower development of ‘Xiang Yun’, the contents of soluble sugar, soluble protein, free proline, and triglycerides were deficient to varying degrees, and the activities of POD, SOD, and MDA were lower. Sequence alignment and qPCR showed that there were several mutations in *EFD1*, *TPD1*, and *DEX1* of ‘Xiang Yun’ compared with ‘Hong Ye’, and the expression levels of these genes were abnormally elevated in the later stages of development. Our results clarified the timing and phenotype of male sterility in ‘Xiang Yun’. This provides solid and valuable information for further research on the molecular mechanism of sterility in ‘Xiang Yun’ and the genetic breeding of crape myrtle.

## 1. Introduction

Male sterility is a common phenomenon in the plant kingdom, where the male organs degenerate or become deformed or underdeveloped during the sexual reproduction process, resulting in the inability to produce normal pollen and leading to sterility [1]. For the plant itself, male sterility is an unfavorable variation, but for breeders, it is a valuable trait resource for utilizing hybrid vigor to improve the quality of offspring. By crossing male sterile lines as the female parent with normal male parent materials, manual emasculation can be eliminated, planting costs can be reduced, and seed purity can be better ensured, improving quality. Plant male sterility has also become an important approach for utilizing hybrid vigor in many agricultural and economic crops and is widely applied in agricultural production [2].

The causes of male sterility are diverse and complex, with different types of plants exhibiting varying methods and timing of male sterility [3,4]. Typical characteristics of sterility include the following: (1) Abnormal anthers: In some plants, the anthers are shriveled and malformed, displaying abnormal colors such as white or brown, or the anthers fail to dehisce normally, with only a few anthers containing aborted pollen or very little normal pollen [5,6,7,8]. (2) Deformed or degenerated stamens: Abortion occurs early in anther development, leading to the stamens withering to mere traces, the stamens developing abnormally into petal-like shapes [9], or the filaments being short and the anthers deformed [10]. (3) Degeneration of the sporogenous sacs: In some plants, the pollen grains develop incompletely, with no sporogenous cells in the anthers; abnormal meiosis of microspores occurs during the pollen mother-cell stage [11]. (4) Microspore degeneration: The development of pollen mother cells, tetrads, tapetum cells, and anther walls is nearly normal, with only the microspores developing abnormally. Depending on the stage of microspore degeneration, it can be further divided into typical abortion, spherical abortion, and dyed abortion [12]. (5) Pollen functional defects: There are morphologically and physiologically normal pollen grains in the anthers, but they cannot germinate under normal conditions and can only germinate after certain specific treatments [13]. These sterility traits are usually regulated by genes, and any abnormality in related genes at any stage of flower development can lead to male sterility in plants. In *Arabidopsis*, the *DEFECTIVE IN EXINE FORMATION1* (*dex1*) mutation disrupts normal primordium development and destroys sporopollenin deposition, causing pollen grains to collapse and, ultimately, affecting the structure of the membrane and sporopollenin deposition [14,15]. The *TAPETUM DETERMINANT1* (*TPD1*) product plays an important role in the differentiation of tapetum cells, and analysis of the male sterile mutant *tpd1* shows that the interruption of TPD1 function leads to the differentiation of tapetum cell precursors into microspore mother cells, with additional microspore mother cells formed in the developing *tpd1* anthers, while the tapetum is absent [16,17].

Plant male sterility often accompanies changes in metabolites and enzyme activities within the body, including sugars, proteins, lipids, proline, malondialdehyde, SOD, etc. In wheat, the anthers of S-type cytoplasmic male sterility (CMS) undergo premature tapetum degeneration, abnormal microspore development, and gradual nutritional deficiency. Moreover, the metabolism and transport of sugars in S-CMS anthers change, thereby reducing the resource supply for microspores and, ultimately, leading to the weakening of male function in wheat S-type cytoplasmic male sterility [18]. In kenaf, the ROS-scavenging ability of the CMS line 722HA is lower than that of 722HB, leading to excessive ROS accumulation, which is specifically manifested as premature tapetum degeneration during microspore development, resulting in abnormal pollen function. This suggests that excessive ROS accumulation may affect the normal development of microspores [19]. In cotton, it has also been found that the accumulation or elimination of ROS leads to abnormal programmed cell death (PCD) of the tapetum, which affects the development of microspores and, ultimately, results in male sterility [20].

*Lagerstroemia indica* belongs to the order Myrtales, family Lythraceae, and genus Lagerstroemia, and is a deciduous or evergreen shrub or small tree. *L. indica* trees have an elegant posture and vibrant floral colors, making them highly ornamental, and they are an important summer ornamental tree species in China favored by people. ‘Xiang Yun’ is a mutant of *L. indica* that was accidentally discovered by our research team in the field and is characterized by flowering without fruiting and having a longer flowering period than normal *L. indica* [21]. Previous research has identified several characteristics of sterility in ‘Xiang Yun’, including sterile and unopened anthers, pollen abortion, and deformed and irregularly arranged filament cells. The stigma features short and loosely arranged papilla cells, a flat style with a narrow stylar canal, loosely arranged epidermal cells, and no distinct nuclei. Additionally, there is no embryo sac cavity in the ovules, and the egg apparatus structure is completely absent in the nucellus. Furthermore, there are significant differences in hormone levels during floral organ differentiation between ‘Xiang Yun’ and ‘Hong Ye’ [22,23,24,25]. However, previous research primarily focused on observing the traits of various organs after sterility had occurred. How to explain the timing of sterility and how metabolites within the body participate in regulation remain unclear. Therefore, elucidating the occurrence period of male sterility in ‘Xiang Yun’ will help reveal the key biological processes controlling pollen development and reproduction. In this study, by observing the pollen development process and measuring related physiological indicators, we compared the differences between ‘Xiang Yun’ and ‘Hong Ye’ and clarified the period and mode of pollen abortion in ‘Xiang Yun’. This provides a valuable natural model for further exploring the molecular and cellular mechanisms of male sterility and offers a scientific basis for precisely improving the reproductive characteristics of crops and other horticultural plants.

## 2. Materials and Methods

### 2.1. Plant Materials

The sterile variety ‘Xiang Yun’ and fertile variety ‘Hong Ye’ were collected from the forest farm of the Hunan Academy of Forestry Sciences, which is located at approximately 113°01′30″ E, 28°06′40″ N. Starting from May 2022, multiple plants with similar growth, normal tree shape, and no pests or diseases were selected. Between 6:00 and 7:00 AM, 5 to 10 fresh flowers from each Lagerstroemia variety were collected and allowed to dehisce naturally, and the pollen was gathered. Some flowers were used for macroscopic morphological observation and pollen viability testing. Additionally, well-grown flowers and fresh buds of different sizes were selected, wrapped in aluminum foil, flash-frozen in liquid nitrogen, and stored at −80 °C for future use.

### 2.2. Morphological Observation of Flower Organs

Flowers and buds from male fertile and sterile varieties of Lagerstroemia with a normal appearance and that were free from pests and diseases were collected separately. Characteristics such as petal color, calyx color, corolla size, and androecium and gynoecium were statistically analyzed and photographed. The transverse and longitudinal diameters of the buds at different stages were measured using a vernier caliper, with five repetitions each. Subsequently, under an optical microscope, the cytological morphological characteristics of buds of different sizes were observed. Buds were dissected on a slide with tweezers, the anthers were picked out and placed on the slide, and the anthers were crushed with a glass rod to gently shake out the microspore cells. A pipette was used to draw up the liquid containing a large number of microspores, which was then dropped onto a clean slide. The slides were stained with acetic acid carmine for 5 min, covered with a coverslip, and photographed. Three buds from each developmental stage were selected for observation, and three fields of view were chosen for each preparation.

### 2.3. Determination of Pollen Germination and Viability

Flowers from the ‘Xiang Yun’ and ‘Hong Ye’ varieties were placed at room temperature until most of the pollen had dispersed. They were then cultured on solid medium (200 g/L sucrose + 5 g/L agar + 150 mg/L boric acid + 20 mg/L calcium chloride) at 25 °C in darkness for about 24 h. Photographs were taken to count the pollen germination rate, which is used to indicate pollen viability.

### 2.4. Anther Anatomical Characteristics and Scanning Electron Microscopic Observation of Pollen

Buds at different developmental stages were fixed in FAA fixative (70% alcohol: glacial acetic acid: formaldehyde: glycerol = 18:1:1:1). After dehydration, the samples were embedded in paraffin and sectioned on a microtome (Leica, Saarbrücken, Germany) to produce 8–12 μm paraffin sections. The sections were stained and immediately observed and photographed under a microscope (Olympus BX51, Hamburg, Germany).

The collected mature pollen grains from the Lagerstroemia plants were dried for 24 h, then coated with gold film for 5 min using a magnetron sputtering coater (Oxford Quorum SC7620, Oxford, UK) with a spray gold current of 10 mA. They were then transferred to a scanning electron microscope (Czech TESCAN MIRA LMS, Brno, Czech Republic), with the acceleration voltage set between 200 eV and 30 keV. Observations and photographs were taken from selected good fields of view. At 300× magnification, the overall shape of the pollen grain group was observed, and at 2000× magnification, the equatorial view of the pollen grain and the pollen germination furrow were examined. Pollen shapes with a P/E value between 1.14 and 2.00 were considered to be prolate spheroids.

### 2.5. Determination of Nutrient Content

Protein content determination was carried out using the Coomassie brilliant blue method: Briefly, accurately weigh 0.25~0.50 g of the sample and add 5 mL of distilled water for grinding. After the sample is ground, centrifuge at low temperature and take the supernatant for later use. Pipette 0.1 mL of the sample extract, then add 5 mL of Coomassie brilliant blue G-250 solution to each centrifuge tube in sequence, mix thoroughly, let it stand for 2 min, and measure the absorbance of the sample at 595 nm. Each sample was measured three times independently, and the protein content was calculated by the difference to a blank control using a standard curve.

Free proline content determination was carried out using a colorimetric method: According to the ratio of weight (g): volume (mL) = 1:9, accurately weigh the mass of buds of different sizes, add 9 times the volume of extraction solution, grind the sample into a homogenate on ice, label it, and then centrifuge it at 3500 rpm for 10 min at low temperature in a centrifuge. After centrifugation, take out the sample and retain the supernatant. According to the instructions in the reagent kit manual (Solarbio Science & Technology Co., Ltd., Beijing, China), add the reagents in sequence and measure the absorbance of each sample. Each sample was measured three times independently.

Soluble sugar content determination was carried out using the anthrone colorimetric method: Weigh approximately 0.1~0.2 g of the sample, add 1 mL of distilled water to grind the sample into a homogenate, boil it in a water bath for 10 min, cool it down, and then centrifuge it at 8000 g at room temperature for 10 min. Dilute with distilled water to 10 mL and shake well until later use. According to the instructions in the reagent kit manual (Solarbio Science & Technology Co., Ltd., Beijing, China), add the reagents in sequence and measure the absorbance of each sample at 620 nm. Each sample was measured three times independently, and a standard curve was plotted to calculate the soluble sugar content.

Triglyceride (TG) content determination was performed using a reagent kit (Solarbio Science & Technology Co., Ltd., Beijing, China). Reagents were added according to the instructions in the reagent kit manual, and the absorbance of each sample was measured at 420 nm. Each sample was measured three times independently.

### 2.6. Determination of Antioxidative Enzyme Activity

Malondialdehyde (MDA) content was determined using the TBA method. Reagents were added in sequence according to the instructions in the reagent kit manual (Solarbio Science & Technology Co., Ltd., Beijing, China), and the absorbance of each sample was measured at 532 nm. The results were calculated, and each sample was measured three times independently.

The activities of catalase (CAT), superoxide dismutase (SOD), and peroxidase (POD) were all determined using reagent kits (Solarbio Science & Technology Co., Ltd., Beijing, China). Reagents were added in sequence according to the instructions in the reagent kit manual, and each sample was measured three times independently.

### 2.7. RNA Extraction, Gene Cloning, and qPCR

Total RNA was extracted from crape myrtle leaves following the instructions of the polysaccharide polyphenol complex plant RNA extraction kit (Aidlab Biotechnologies Co., Ltd., Beijing, China), and the quality of the RNA was assessed via agarose gel electrophoresis. Non-degraded RNA was then used to synthesize cDNA with a reverse transcription reagent (Vazyme Biotechnologies Co., Ltd., Nanjing, China) and subsequently diluted. Primers were designed at both ends of the CDS sequence of *L. indica* (Appendix A). After PCR amplification, the products were purified, and the gene sizes were determined by agarose gel electrophoresis. The amplified fragments were then ligated into the pTOPO-Blunt cloning vector (Aidlab Biotechnologies Co., Ltd., Beijing, China), transformed into DH5α cells, and finally sent to BGI Genomics (Guangzhou, China) for sequencing to obtain the successfully cloned genes.

For the qPCR method, fertile ‘Hong Ye’ was selected as the sample control, with the *Li18S* gene serving as the internal reference gene, and testing of each sample was replicated three times. After diluting the cDNA by 4×, 16×, 64×, and 256×, amplification was carried out to establish a standard curve and to calculate the amplification efficiency. The reaction system and amplification program were set according to the manufacturer’s instructions, with the standard program chosen and data analysis performed using the 2^−ΔΔCt^ method [26].

### 2.8. Data Processing

All experimental data were obtained from three replicates and reported as the mean ± standard deviation (SD). GraphPad Prism 10 was used for two-Way ANOVA multivariate variance analysis and bar graph plotting.

## 3. Results

### 3.1. Morphological Analysis of Floral Organs of ‘Xiang Yun’

The morphological characteristics of the ‘Xiang Yun’ floral organs are shown in Figure 1. Compared with the fertile ‘Hong Ye’, ‘Xiang Yun’ has a slightly larger corolla, measuring 5.32 × 4.95 cm, with 20~30 short stamens, and the petals are light pink. The exterior of the buds of different sizes is mainly light green. ‘Hong Ye’ has purple-red petals, a corolla size of 4.20 × 3.87 cm with 20~40 short stamens, and the bud epidermis is mainly dark red with light red filaments and styles. Both varieties have six petals and six long stamens. Further measurement of bud sizes at different stages (Appendix A) shows that the longitudinal and transverse diameters of the ‘Xiang Yun’ buds gradually increase with development. Compared with ‘Hong Ye’, there are no significant differences in the first three stages of development, but in the later stages, the size of the ‘Xiang Yun’ buds is larger than that of the ‘Hong Ye’ buds. There are no obvious differences in floral morphology between the two, and ‘Xiang Yun’ does not show obvious sterility characteristics in its external phenotype.

### 3.2. Analysis of Pollen Development of ‘Xiang Yun’

Based on the development process of microspores, they are divided into five stages (Figure 2a1–b5), namely the pollen mother-cell stage (S1), the tetrad stage (S2), the early uninucleate stage (S3), the late uninucleate stage (S4), and the pollen maturity stage (S5). As the microspores develop, the longitudinal and transverse diameters of the flower buds of ‘Hong Ye’ and ‘Xiang Yun’ also continue to increase. In the last three stages, the longitudinal and transverse diameters of the flower buds of ‘Xiang Yun’ are both larger than those of ‘Hong Ye’, but the difference between the two is not significant.

Through the comparison of pollen grain size and shape, it was found that the polar axis length, equatorial axis length, pollen size, germination groove length, and groove ridge width of the long- and short-stamen pollen grains of ‘Xiang Yun’ are all smaller than those of ‘Hong Ye’ pollen grains. According to Appendix A, the polar axis lengths of the long- and short-stamen pollen grains of ‘Xiang Yun’ are 32.64 μm and 31.66 μm, respectively; the equatorial axis lengths of the long- and short-stamen pollen grains are 19.21 μm and 22.24 μm, respectively; the sizes of the long- and short-stamen pollen grains are 626.30 μm² and 704.05 μm², respectively; the germination groove lengths of the long- and short-stamen pollen grains are 22.22 μm and 20.21 μm, respectively; and the groove ridge widths of the long- and short-stamen pollen grains are 5.72 μm and 6.33 μm, respectively. In contrast, the size of the short-stamen pollen grains of ‘Hong Ye’ is 1267.10 μm². From the perspective of pollen shape, both belong to prolate spheroid pollen grains, and there are significant differences in different traits of the long- and short-stamen pollen grains between the two varieties.

Scanning electron microscopic observations of pollen and anthers from two cultivars are illustrated in Figure 2c1–f3. In terms of the pollen population, for both the long- and short-stamen pollen of ‘Xiang Yun’, the majority of the pollen grains appear shrunken and irregular. Through measurement and statistical analysis, the external morphological index data of ‘Xiang Yun’ pollen grains all have smaller values than those of ‘Hong Ye’, including the polar axis length, equatorial axis length, and germination groove length (Appendix A). It can be observed through the short-stamen pollen grains that both cultivars have numerous germination pores on the surface of the pollen. However, on the long-stamen pollen grains, it is evident that the germination groove ridges and poles of ‘Hong Ye’ pollen grains are smoother, and the overall shape is plumper, while the surface of ‘Xiang Yun’ pollen grains is adorned with dense brain-like folds (Figure 2c3–f3).

To observe the viability of ‘Xiang Yun’ pollen grains, the pollen germination rates of the sterile and fertile varieties were measured, and in vitro pollen-tube germination experiments were conducted using solid culture medium. As can be seen from Figure 2g,h, most of the pollen of the fertile variety ‘Hong Ye’ can germinate, with a pollen germination rate of 92.76% after statistical analysis. However, in the culture medium of the ‘Xiang Yun’ pollen, it can be seen that the pollen hardly germinates, indicating that the vitality of ‘Xiang Yun’ pollen grains is low.

The development of pollen grains at different periods for ‘Hong Ye’ and ‘Xiang Yun’ is shown in Figure 3. Tissue sections of the anthers from both cultivars at different periods were made to observe the development of the pollen grain cells. At stage S1, the microspores of both cultivars began meiosis, and they were morphologically similar, with no obvious differences observed (Figure 3a1,b1). At stage S2, compared with ‘Hong Ye’, the tetrads in ‘Xiang Yun’ wrapped more tightly around the microspores (Figure 3a2,b2). The interior of the ‘Xiang Yun’ anther began to show irregular, incomplete tetrad structures and abnormally developing microspores. At stage S3, the microspores of ‘Hong Ye’ appeared round (Figure 3a3), while those of ‘Xiang Yun’ were angular (Figure 3b3). At stage S4, the microspores of ‘Hong Ye’ became vacuolated and formed a regular outer wall foundation (Figure 3a4), whereas the microspores of ‘Xiang Yun’ began to degenerate (Figure 3b4). At stage S5, with the cracking of the anthers, the microspores of ‘Hong Ye’ were released from the locules (Figure 3a5). However, the microspores of ‘Xiang Yun’ completely aborted (Figure 3b5). These results indicate that abnormalities in microspore development began to appear from the S2 stage, and that the callose walls between microspores in the tetrads of ‘Xiang Yun’ were thicker than those of ‘Hong Ye’ (Figure 3c,d).

The observation results of lipid substances are shown in Figure 3e1–f4. During the late uninucleate stage and pollen maturity stage, the mature pollen grains of ‘Xiang Yun’ are fewer and stain lighter, indicating that ‘Xiang Yun’ accumulates fewer lipid substances within its pollen during the maturity stage. In contrast, the staining of the pollen of the fertile variety ‘Hong Ye’ at the same stage clearly shows that the pollen grains are well stained, with almost all the pollen grains being dyed as dark blue–black granular substances, indicating that the lipid substances in the pollen of ‘Hong Ye’ continue to accumulate with the continuous development of the anthers, reaching a maximum value at the pollen maturity stage.

### 3.3. Determination of Nutrient Content in the ‘Xiang Yun’ Flower During Its Development

The changes in nutrient content during the development of ‘Xiang Yun’ flowers are shown in Figure 4. The trend for the soluble protein content in both ‘Xiang Yun’ and ‘Hong Ye‘ is basically consistent across the five stages of bud development, with both continuously increasing with the development of the buds and accumulating to their maximum values during the pollen maturity stage. However, the soluble protein content of ‘Hong Ye’ is higher than that of ‘Xiang Yun’ at all stages of microspore development, reaching 97.987 mg/g during the pollen maturity stage. The soluble protein content of ‘Xiang Yun’ is lowest at the S2 stage, only 18.798 mg/g, and although it increases to 61.718 mg/g by the S5 stage, it remains significantly lower than that of ‘Hong Ye’. The content of free proline generally shows an upward trend with the development of microspores, with little change in the first three stages of development. At the S4 stage, there is a significant difference in the free proline content between ‘Xiang Yun’ and ‘Hong Ye’, with that of ‘Xiang Yun’, at 62.827 μg/g, being only half that of ‘Hong Ye’. The content of triglycerides does not change much during the development of ‘Xiang Yun’, while the triglyceride content of ‘Hong Ye’ continuously accumulates with the development of the buds, reaching a maximum value of 1.134 mg/g at the S5 stage. At the same time, the triglyceride content of ‘Xiang Yun’ is only 0.333 mg/g, which is significantly lower than that of ‘Hong Ye’. Throughout the entire bud development process, the triglyceride content of ‘Hong Ye’ is consistently higher than that of ‘Xiang Yun’, which is consistent with the results of previous lipid slice observations. At 11.236 mg/g, the soluble sugar content of ‘Xiang Yun’ is highest during the tetrad stage and gradually decreases as the bud continues to develop, reaching 4.009 mg/g at the S5 stage. The soluble sugar content of ‘Hong Ye’ generally shows a trend of first decreasing and then increasing, and it continuously accumulates in the later stages.

### 3.4. Determination of Reactive Oxygen Species During the Development of the ‘Xiang Yun’ Flower

The changes in reactive oxygen species activity during the development of ‘Xiang Yun’ flowers are shown in Figure 5. As the buds develop, the POD activity of ‘Xiang Yun’ shows a trend of first increasing and then decreasing, while the POD activity of ‘Hong Ye’ shows a gradual decrease, and the POD activity of ‘Hong Ye’ is always higher than that of ‘Xiang Yun’, with significant differences in the first three stages. The SOD activity of ‘Xiang Yun’ is significantly lower than that of ‘Hong Ye’, reaching a minimum value of 378.568 U/g in the S5 stage. The SOD activity of ‘Hong Ye’ gradually increases in the first three stages, reaching a maximum value of 910.951 U/g in the S3 stage, and also shows a downward trend in the later stages of development. Among them, the difference in SOD activity between the two is significant in the S3 stage, with the SOD activity of ‘Hong Ye’ being 1.57 times that of ‘Xiang Yun’. MDA content measurement shows that throughout the development period, the malondialdehyde content in the buds of ‘Hong Ye’ is always higher than that of ‘Xiang Yun’, with a significant difference in the S4 stage. The malondialdehyde content of ‘Hong Ye’ decreases in the first three stages and significantly increases in the S4 stage, reaching a maximum value of 110.147 nmol/g, while the malondialdehyde content of ‘Xiang Yun’ is only 31.918 nmol/g during the same period. The malondialdehyde content of ‘Xiang Yun’ remains at a relatively low level throughout the development process, showing an overall trend of first slowly rising and then slowly falling. The change in catalase activity shows that the CAT activity of ‘Hong Ye’ shows a maximum value of 43.693 U/g in the S1 stage, with an overall trend of first decreasing and then slowly rising but no obvious trend in the other development periods. The CAT activity of ‘Xiang Yun’, on the other hand, shows a maximum value of 58.913 U/g in the S2 stage, which is 2.65 times that of ‘Hong Ye’ for the same period, and rapidly decreases in the S3 stage, showing an overall upward trend in the later stages of development, which is the same as the change in the CAT activity of ‘Hong Ye’.

### 3.5. Sequence Comparison and Expression Analysis of Genes Related to Pollen Development

To explore the genetic differences between sterile and fertile crape myrtles, combined with the aforementioned research and results on the cytological and physiological characteristics, three genes known to be associated with male sterility, namely *EXINE FORMATION DEFECT* (*EFD1)*, *TPD1*, and *DEX1*, were selected for sequence alignment and quantitative analysis. As can be seen from Figure 6a, compared with ‘Hong Ye’, ‘Xiang Yun’ exhibits some single nucleotide polymorphisms and base mutations in these genes. In ‘Xiang Yun’, *EFD1* mainly contains transition mutations, while *TPD1* and *DEX1* have a small number of transversion mutations, which may lead to missense mutations in the gene sequence, thereby causing differences between ‘Xiang Yun’ and ‘Hong Ye’. Subsequent quantitative analysis of flowers at different stages showed that the expression levels of the three genes in ‘Xiang Yun’ were generally consistent across the different developmental stages, reaching maximum expression levels during the S4 stage, and these differences were significant compared with ‘Hong Ye’ (Figure 6b). *TPD1* and *EFD1* had lower expression levels in ‘Hong Ye’, with significantly higher expression only during the S2 stage compared with ‘Xiang Yun’. The expression level of *DEX1* in ‘Hong Ye’ fluctuated less during development, while in ‘Xiang Yun’, the expression level of *DEX1* showed an upward trend followed by a decline, with maximum expression during the S4 stage. These results indicate that these abnormally increased gene expression levels in ‘Xiang Yun’ in the later stages of flower development may affect the normal formation of pollen.

## 4. Discussion

Pollen development is a crucial biological process in flowering plants, which starts with the differentiation of anther sporogenous cells and is followed by several key stages, including the formation of pollen mother cells, the progression of meiosis, the formation and construction of the pollen wall, the degeneration of the tapetum, and mitosis of the pollen, until the pollen reaches maturity. Mature pollen grains are naturally released when the anthers dehisce [27,28]. The main reasons for pollen abortion include abnormal formation of the pollen sacs, abnormal meiosis, abnormal callose, abnormal tapetum, abnormal pollen wall, and abnormal anther dehiscence, among others [29,30,31]. These factors causing pollen abortion are not entirely independent; many cases of pollen abortion are caused by a combination of multiple factors. This study observed the process of ‘Xiang Yun’ floral organs and pollen development from a morphological perspective. ‘Xiang Yun’ floral organs are normal, and the amount of pollen is normal, but the pollen vitality is low. This phenomenon is similar to that of kenaf [19]. Cytological observations indicate that ‘Xiang Yun’ pollen begins to show abnormalities during meiosis, with abnormal chromosome division leading to the appearance of many irregular tetrad structures. In the later stages of development, apoptosis of the nucleus occurs in late uninucleate pollen grains. By the maturity stage, inactive pollen grains that are hollow, shriveled, and uneven in size are formed. Therefore, abnormalities during meiosis and abnormal development of the tapetum may be important reasons for the abortion of ‘Xiang Yun’ microspores. This abnormality is very similar to the phenotypes reported in many studies related to pollen abortion, such as the homozygous mutation of the *Arabidopsis* mps1 gene, which produces unevenly sized microspores during anther development, most of which abort [32]. In the monocot plant rice, after gene mutation of the SDS cell cycle protein, many chromosomes in the pollen mother cells remain uninucleate during the meiotic prophase and cannot normally pair to form binucleate cells, ultimately forming aborted pollen grains of varying sizes [33].

Pollen maturity is accompanied by changes in metabolite content, with the levels of free proline, sugars, proteins, and lipids being closely related to pollen fertility [34,35]. Free proline provides an important energy source and ammonia for pollen germination and pollen-tube elongation; a lack of proline may affect the structural changes of metabolites, leading to microspore abortion. Sugars are raw materials for the synthesis of anther development substances, providing energy for microspore development; a deficiency in sugar substances can also lead to reduced pollen vitality or male sterility [36]. Proteins are nutrients accumulated during the microspore development process, and metabolic disorders causing nutritional imbalance are the main reasons for abortion [37,38]. The transport of lipids in the anther is crucial for the formation of the pollen wall and pollen grains [39]. In this study, the contents of soluble sugars, free proline, proteins, and triglycerides in ‘Xiang Yun’ were generally lower than those in ‘Hong Ye’. Among them, the contents of soluble sugars and free proline in ‘Xiang Yun’ were higher than those in ‘Hong Ye’ in the early stages of development, but as the flower buds grew and developed, the content of soluble sugars in ‘Xiang Yun’ gradually decreased, and the contents of free proline, proteins, and lipids remained at a low level with little change. This indicates that the deficiency of metabolites in ‘Xiang Yun’ during flower development leads to reduced fertility of the microspores, which is one of the reasons for male sterility in crape myrtle flowers. This result is similar to changes in the endogenous substance content in other male sterile plants, such as the low level of proline content in the anthers of sterile lilies at three developmental stages [40]. Thermosensitive nuclear male sterile wheat also exhibits characteristics of pollen starvation and impaired starch accumulation [41].

ROS are important signaling molecules involved in many biochemical processes of normal plant growth and development [42]. However, excessive ROS can damage protein structures and act as toxins that kill cells [43]. Therefore, plants must counteract the oxidative stress caused by excessive ROS through an effective scavenging system to avoid oxygen-induced damage. Antioxidative enzymes such as POD, SOD, and CAT can help clear excess amounts of aerobic free radicals that accumulate in the plant body, preventing the excessive accumulation of reactive oxygen species, which can cause membrane damage [44]. These three enzymes work together synergistically to maintain a stable level of free radicals within the plant body, ensuring that the level of free radicals is in homeostasis, thereby preventing the adverse effects on plant physiological and biochemical processes caused by free radicals. In this study, the activities of POD and SOD in ‘Xiang Yun’ were lower than those in ‘Hong Ye’ at multiple stages, and the content fluctuated less with the development of the flower buds. This result is similar to that of kenaf but inconsistent with that of chili peppers, possibly because the type of sterility that occurs in ‘Xiang Yun’ is different from that in chili peppers [45,46]. CAT activity showed a trend of decreasing first and then increasing, and at stage S2, the CAT activity of ‘Xiang Yun’ was significantly higher than that of ‘Hong Ye’. We speculate that there may be excessive ROS in the body of ‘Xiang Yun’ during stage S2, leading to abnormally high CAT activity. Membrane lipid stability is a basic condition for maintaining the normal life activities of cells. Malondialdehyde, as an important indicator of plant lipid peroxidation, can cause cell membrane structure damage or injury when accumulated in large amounts during microspore development, thereby causing chaos in the physiological and biochemical metabolism within the cells [47]. The results of this study show that the MDA content in ‘Xiang Yun’ is higher than that in ‘Hong Ye’, which may be one reason for the abnormalities in the microspores of ‘Xiang Yun’.

The development of the anther, from the origin of the stamen primordium cells to anther dehiscence and the release of mature pollen, is regulated by genes. Any abnormality in any link can lead to abnormal anther development, resulting in sterile pollen. The pollen wall is crucial for male fertility, and defects in pollen wall development are a common cause of male sterility in plants [48]. At the tetrad stage, the formation of primexine is essential for the normal development of exine. Previous studies on several mutants with primexine formation defects (*efd1*, *dex1*, *nef1*, *rpg1*, and *npu*) have revealed that correct primexine formation is a key factor in exine development [49,50]. In this experiment, three amino acids in *EFD1* of ‘Xiang Yun’ were found to be mutated, and *TPD1* and *DEX1* each had two mutated amino acids. These mutated amino acids can effectively enhance the plant‘s stress resistance. The qPCR results show that the expression levels of *EFD1*, *TPD1*, and *DEX1* in ‘Hong Ye’ are relatively stable. However, we noticed that *EFD1* and *TPD1* are strongly expressed during the S2 stage. Combining the functions of these two genes in *Arabidopsis*, it is known that *EFD1* and *TPD1* have high expression levels during microsporogenesis and are involved in the formation of the tapetum [51]. The expression levels of *EFD1* and *TPD1* in ‘Xiang Yun’ are extremely low, suggesting that the impaired development of the tapetum in ‘Xiang Yun’ leads to its reduced gene expression. During the S4 stage, there are significant differences in the expression levels of six genes in ‘Xiang Yun’ compared with ‘Hong Ye’. We speculate that these genes may have a feedback mechanism in the late stage of pollen development in ‘Xiang Yun’, where abnormal increases in gene expression levels lead to decreased pollen viability and quantity.

## 5. Conclusions

In this study, we explored the mechanism of male sterility in ‘Xiang Yun’ from the morphological, cytological, and physiological–biochemical aspects. Microscopic observations revealed that the anthers of ‘Xiang Yun’ dispersed pollen normally, but most pollen grains were inviable and unable to germinate normally both in vivo and in vitro, with uneven sizes, unclear grooves, and insufficiently full morphology. By detecting different physiological indicators, it was found that during the flower development of ‘Xiang Yun’, the contents of soluble sugar, soluble protein, free proline, and triglycerides showed varying degrees of deficiency, affecting the normal development of pollen grains and anthers, which is one of the inducements for the plant to develop male sterility. Low levels of POD and SOD disrupted the balance of reactive oxygen species, causing an excessive protective response, leading to membrane stability disorder and, thus, reducing fertility. Sequence alignment and qPCR results showed that the gene sequences of *EFD1*, *TPD1*, and *DEX1* in ‘Xiang Yun’ differed from those in ‘Hong Ye’, and that the expression levels of these genes were abnormally elevated in the later stages of development. Our results clarified the timing and phenotype of male sterility occurrence in ‘Xiang Yun’. The research is expected to provide new insights into the variety improvement of *L. indica* and other horticultural plants.

## Figures and Tables

**Figure 1 plants-13-03043-f001:**
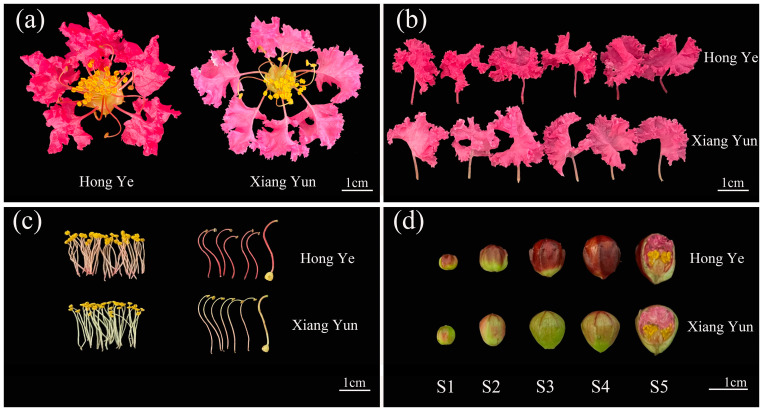
Comparison of the floral morphology between ‘Hong Ye’ and ‘Xiang Yun’. (**a**) Overall flower morphology; (**b**) Petals; (**c**) Long and short stamens and stigma; (**d**) Buds of different sizes; S1–S5 represent different developmental stages. Scale bar = 1 cm.

**Figure 2 plants-13-03043-f002:**
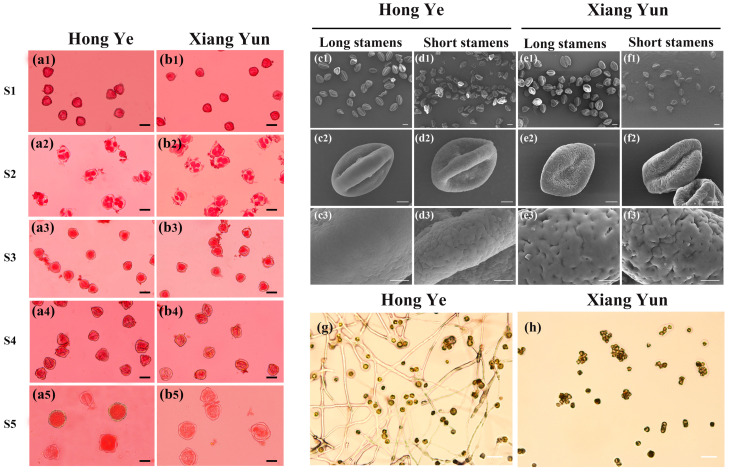
Comparison of pollen from ‘Hong Ye’ and ‘Xiang Yun’ at different developmental stages. (**a1**–**b5**) Pollen microspores at different developmental stages; scale bar = 20 μm. (**c1**–**f3**) Scanning of pollen morphology in long and short stamens; scale bar = 10 μm. (**g**,**h**) Pollen vitality test of ‘Hong Ye’ and ‘Xiang Yun’; scale bar = 1 mm.

**Figure 3 plants-13-03043-f003:**
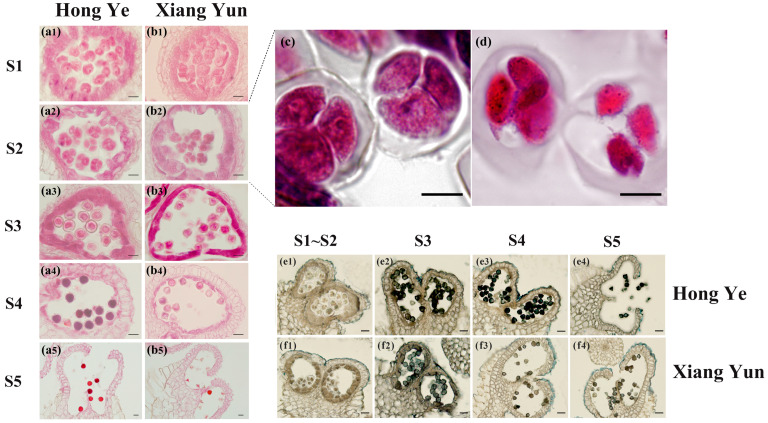
A comparison of the anthers of ‘Hong Ye’ and ‘Xiang Yun’ at different developmental stages. (**a1**–**b5**) Observation of anther sections at different developmental stages; scale bar = 30 μm. (**c**,**d**) Observation of tetrad morphology in ‘Hong Ye’ and ‘Xiang Yun’; scale bar = 30 μm. (**e1**–**f4**) Observation of lipid staining in anthers at different developmental stages; scale bar = 50 μm.

**Figure 4 plants-13-03043-f004:**
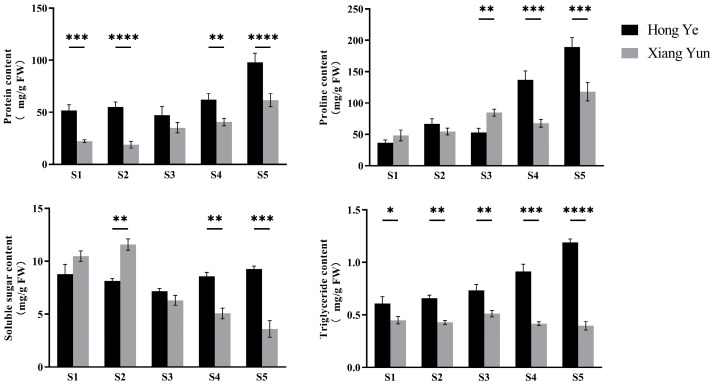
Variations in the nutrient content of crape myrtle buds during different stages. The values represent the mean of three independent replicates, with error bars indicating the standard deviation. *, **, ***, and **** denote significant differences at the *p* < 0.05, *p* < 0.01, *p* < 0.001, and *p* < 0.0001 levels, respectively, as determined by two-way ANOVA.

**Figure 5 plants-13-03043-f005:**
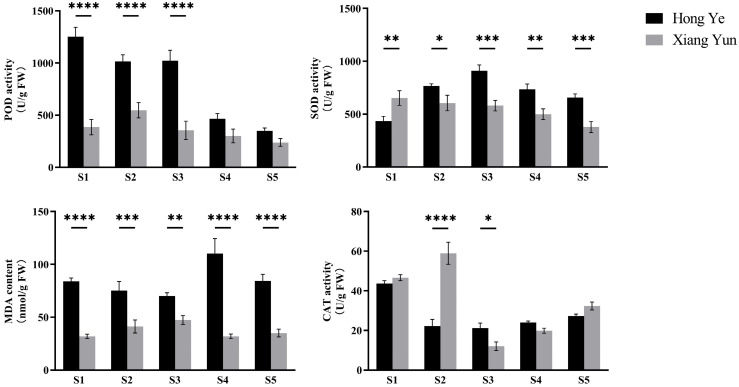
Variations in the metabolism of reactive oxygen species in crape myrtle buds during different periods. The values are the average of three independent replicates, with error bars indicating the standard deviation. *, **, ***, and **** denote significant differences at the *p* < 0.05, *p* < 0.01, *p* < 0.001, and *p* < 0.0001 levels, respectively, as determined by two-way ANOVA.

**Figure 6 plants-13-03043-f006:**
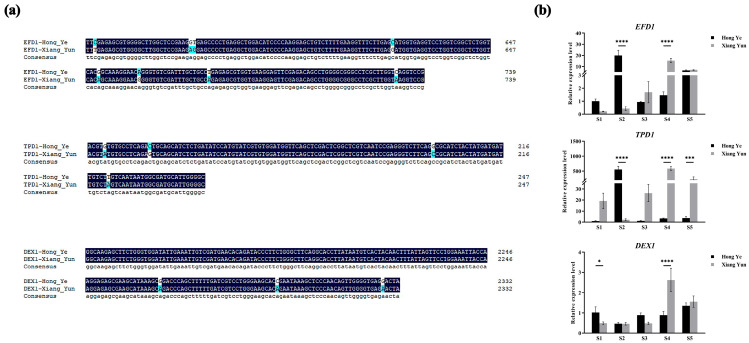
Differences in pollen development genes between ‘Xiang Yun’ and ‘Hong Ye’. (**a**) Sequence alignment of pollen development-related genes. (**b**) Expression levels of pollen development-related genes at different stages of bud development. The values are the average of three independent replicates, with error bars indicating the standard deviation. *, ***, and **** denote significant differences at the *p* < 0.05, *p* < 0.01, *p* < 0.001, and *p* < 0.0001 levels, respectively, as determined by two-way ANOVA.

## Data Availability

The original contributions presented in the study are included in the article/Appendix A, further inquiries can be directed to the corresponding author.

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
