# Peer review of "Floral Developmental Morphology and Biochemical Characteristics of Male Sterile Mutants of *Lagerstroemia indica"

_plants, 2024, doi:10.3390/plants13213043_

Round 1

Reviewer 1 Report

Comments and Suggestions for Authors

Introduction

  1. Strengthen the rationale for studying 'Xiang Yun'
    • Clearly explain the importance of this mutant for understanding male sterility mechanisms.
    • Highlight how this research contributes to both basic plant science and horticultural applications.
  2. Emphasize broader implications
    • Discuss how findings from 'Xiang Yun' could inform our understanding of plant reproduction in general.
    • Explore potential applications in breeding programs beyond Lagerstroemia indica.
  3. Define technical terms
    • Provide brief explanations for specialized terms like "S2 stage", "EFD1", and "TPD1" upon first use.
    • Consider adding a glossary if many technical terms are used throughout the paper.
  4. Expand on previous work
    • Provide a more comprehensive summary of existing research on 'Xiang Yun'.
    • Clearly state the limitations of previous studies and how this current work addresses those gaps.

Materials and Methods

     5. Clarify experimental design

  • Specify the number of biological replicates for each experiment.
  • Describe how many plants were sampled for each variety and how they were selected.
  1. Enhance qPCR methodology description
    • Detail the normalization process used in gene expression analysis.
    • Explain how primer efficiency was determined and considered in the analysis.
    • If multiple reference genes were used, describe how they were selected and validated.

Discussion

     7. Contextualize findings

  • Expand the comparison of results to other ornamental species exhibiting male sterility.
  • Discuss how these findings contribute to the broader understanding of plant reproductive biology.

  1. Explore applications
    • Discuss potential applications of this research in breeding programs.
    • Consider how the findings might inform genetic engineering approaches in Lagerstroemia indica or related species.

Reviewer 2 Report

Comments and Suggestions for Authors

The subject of the article entitled “Floral Developmental Morphology and Biochemical Characteristics of Male Sterile Mutants in Lagerstroemia indica” written by Fuyuan Deng, Liushu Lu, Lu Li, JingYang, Yi Chen, Huijie Zeng,Yongxin Li and Zhongquan Qiao is connected to the structure of the flower elements as well the biochemical properties of crape myrtle, which is a commonly known as an ornamental species. Lagerstroemia indica is native to China but widespread throughout the and planted in different countries like the USA or Australia. The authors of the article concentrate on the research of a sterile variety and compare the mutant to the fertile variety. 

            In a (relatively) long introduction (part 1. “Introduction”), the authors emphasized the benefits that can appear after using sterile plants in agriculture. They focused not only on the sterility phenomenon but also on the physiological aspects that come with it. The 22 literature items cited in the “Introduction” of the paper gives the quite extensive and appropriate background to the research presented in the further parts of reviewed paper.

In the next, second part of the publication, entitled “2. Material and Methods” the authors give the data connected to the place of plant growth, material collection and in the next subsections describe the methods used throughout the study. These methods were chosen and used properly and adequately to the planned goals of the research.

The third part (“3. Results”) presents the results obtained by authors and is summarized in six detailed figures. This part of the paper is extensive. It contains numerous photographs and diagrams which contain extensive data. The research documentation and the description of the obtained results are presented in details and adequate to the requirements of such a section.

In part “4. Discussion” the results obtained by the authors are thoroughly analyzed and discussed in a manner appropriate for scientific publications. The results obtained by the authors are compared to the appropriate literature data.

The main results of the research are formulated and summarized in the conclusions given in part 5 (5. “Conclusion”). Presented conclusions are concise and adequate to the conducted research.

In summary, the paper is concrete, properly conducted and well written. I would not like to go into details about the work because it will repeat the content contained therein. To sum up, I find this paper interesting and it is suitable for publishing in “Plants”.

Author Response

Thank you very much for taking the time to review this manuscript. Your positive feedback and encouraging remarks on the manuscript are what drive us to continually enhance our work. We have made modifications to certain parts of the manuscript, which you can review in the newly uploaded version and provide further feedback. Thank you once again for your affirmation of our efforts.

Reviewer 3 Report

Comments and Suggestions for Authors

This paper studies in detail the possible causes of male sterility in the species Lagerstroemia indica. I congratulate the authors for the breadth of the study undertaken, but I would like to suggest some modifications to further enhance it.

Materials and Methods

Authors consider abnormal anthers white, why?. This color is common in same cases when direct sunlight does not occur.

Bud sizes at different stages are called S1, S2… but the same names are selected for development process of microspores. These two evolutionary processes are coincident?

Results

The work of evolutionary study of pollen has been very thorough, however Figure 2, detracts from all this since it is impossible to appreciate the anatomical changes in section a-j. Same comment for Figure 3.

Discussion

420.- Authors said: Cytological observations indicate that 'Xiang Yun' pollen begins to show abnormalities during meiosis, with abnormal chromosome division leading to the appearance of many irregular tetrad structures and apoptosis of the nucleus occurs in late uninucleate pollen grains.These anomalies were not showed in results
